# Spatiotemporal dynamics of multi-vesicular release is determined by heterogeneity of release sites within central synapses

Dario Maschi, Vitaly A Klyachko*

Department of Cell Biology and Physiology, Washington University School of Medicine, St. Louis, United States

**Abstract** A synaptic active zone (AZ) can release multiple vesicles in response to an action potential. This multi-vesicular release (MVR) occurs at most synapses, but its spatiotemporal properties are unknown. Nanoscale-resolution detection of individual release events in hippocampal synapses revealed unprecedented heterogeneity among vesicle release sites within a single AZ, with a gradient of release probability decreasing from AZ center to periphery. Parallel to this organization, MVR events preferentially overlap with uni-vesicular release (UVR) events at sites closer to an AZ center. Pairs of fusion events comprising MVR are also not perfectly synchronized, and the earlier event tends to occur closer to AZ center. The spatial features of release sites and MVR events are similarly tightened by buffering intracellular calcium. These observations revealed a marked heterogeneity of release site properties within individual AZs, which determines the spatiotemporal features of MVR events and is controlled, in part, by non-uniform calcium elevation across the AZ.

*For correspondence:
klyachko@wustl.edu

**Competing interests:** The authors declare that no competing interests exist.

## Introduction

Information transmission in the brain critically relies on the number of vesicles released in response to each action potential. Not surprisingly, there have been several major efforts to model this process in recent years (*Neher, 2010*; *Pan and Zucker, 2009*; *Rotman et al., 2011*). Although the initial hypothesis was that, at most, only a single vesicle is released by a given synapse in response to an action potential (i.e., uni-vesicular release (UVR)), we now know that two or more vesicles can be released in response to a single action potential, even within the same synaptic bouton (*Rudolph et al., 2015*). This phenomenon has been named multi-vesicular release (MVR). Indeed, MVR is a ubiquitous release mechanism, occurring at both small and large central synapses throughout the brain (*Auger et al., 1998*; *Christie and Jahr, 2006*; *Foster et al., 2005*; *Huang et al., 2010*; *Leitz and Kavalali, 2011*; *Malagon et al., 2016*; *Oertner et al., 2002*; *Rudolph et al., 2011*; *Singer et al., 2004*; *Taschenberger et al., 2002*; *Tong and Jahr, 1994*; *Wadiche and Jahr, 2001*). Because the vast majority of small central synapses contain only a single active zone (AZ) (*Schikorski and Stevens, 1997*; *Schikorski and Stevens, 1999*), it follows that individual AZs must be capable of supporting both UVR and MVR.

MVR has been suggested to serve a wide range of functions, including enhancing synaptic reliability, controlling synaptic integration, enhancing efficient information transmission by complex spikes, and inducing synaptic plasticity (*Rudolph et al., 2015*). However, despite the prevalence and functional significance of MVR, the regulatory mechanisms and spatiotemporal organization of MVR events within individual synaptic AZs are poorly understood. Indeed, we know relatively little about the functional organization of vesicle release within the AZ in general—be it related to UVR or MVR

events—although findings of nanoscale co-clustering of presynaptic docking factors and vesicle release machinery (*Bademosi et al., 2016*; *Glebov et al., 2017*; *Tang et al., 2016*; *Weyhersmüller et al., 2011*) have underscored the idea that vesicles are released from relatively stable 'release sites'. Progress toward this end is hampered by the fact that the AZ is extremely small and thus beyond the resolution limits of conventional experimental approaches (*Schikorski and Stevens, 1997*; *Schikorski and Stevens, 1999*).

Recently, we were able to overcome this limitation by developing a nanoscale imaging modality that is capable of resolving the locations of individual vesicle release events in active hippocampal synapses in culture with ~27 nm precision (*Maschi and Klyachko, 2017*). Using these tools, we uncovered the presence of multiple, distinct release sites within individual AZs, at which vesicle fusion occurs repeatedly in response to action potentials. Having demonstrated our ability to identify individual release sites reliably, we now ask a series of questions related to the organization and regulation of these sites within a single AZ: do all sites support vesicle fusion involved in both UVR and MVR events? What controls the relative probability that a site is involved in an MVR event? And, at a more fundamental level: is the probability of release uniform across all sites?

To address these questions, we applied nanoscale imaging tools to detect and study the organizational principles of UVR and MVR events at individual hippocampal synapses in dissociated neuronal cultures. Our results reveal that release site characteristics are highly heterogeneous, even within a single AZ. Specifically, we find that the closer a site is to the center of the AZ, the higher its release probability. Interestingly, this gradient of release site properties also underlies the spatial and temporal organization of vesicle release involved in MVR events. This gradient of release site properties and the spatial features of MVR are also similarly affected by buffering intracellular calcium. Together, our analyses suggest a new level of functional organization of the AZ that is determined by the heterogeneous landscape of release site properties. The spatiotemporal organization of MVR is shaped by the gradient of release site properties across individual AZs and depends, in part, on the non-uniform elevation of calcium across the AZ following an action potential.

## Results

### Detection of MVR events

To detect MVR events and resolve their locations in hippocampal synapses, we took advantage of a nanoscale imaging approach that we recently developed (*Maschi and Klyachko, 2017*), combined with the use of a pH-sensitive indicator, pHluorin, which was targeted to the vesicle lumen via vGlut1 (vGlut1-pHluorin) (*Balaji and Ryan, 2007*; *Leitz and Kavalali, 2011*; *Voglmaier et al., 2006*). vGlut1-pHluorin was expressed in cultures of excitatory hippocampal neurons using lentiviral infection at DIV3. Imaging was then performed at DIV16–19 at 37°C. Release events were evoked using 1 Hz stimulation for 120 s (*Figure 1A*). Robust detection of individual release events was achieved at 40 ms/frame rate throughout the observation time period.

Using a hierarchical clustering algorithm, as well as simulations and statistical considerations, we have previously determined that release events are not randomly distributed throughout the AZ, but occur in a set of defined and repeatedly reused release sites within the AZs (*Maschi and Klyachko, 2017*). Our current data are consistent with that finding (*Figure 1A* and see 'Materials and methods' for details). In addition, visual examination of these recordings revealed a subset of events that involve the simultaneous fusion of two vesicles in the same bouton following a single action potential (*Figure 1A,B*). To identify these double fusion events (i.e., MVR) automatically and to determine their precise spatial locations, we used a well-established mixture-model fitting approach with two fixed-width Gaussians to approximate the point spread function (PSF)-like images of each vesicle (*Jaqaman et al., 2008*; *Thomann et al., 2003*). We previously used a conceptually similar fitting approach to localize individual non-overlapping UVR events, achieving ~27 nm precision (*Maschi and Klyachko, 2017*).

Here, when studying instances of MVR, we found that although the distance between the two fusion events comprising MVR varied widely, shorter separation distances were more frequently observed (*Figure 1C*). Over 90% of fusion event pairs that are involved in MVR were separated by less than 600 nm. The chances of misidentifying two events in the neighboring boutons as occurring

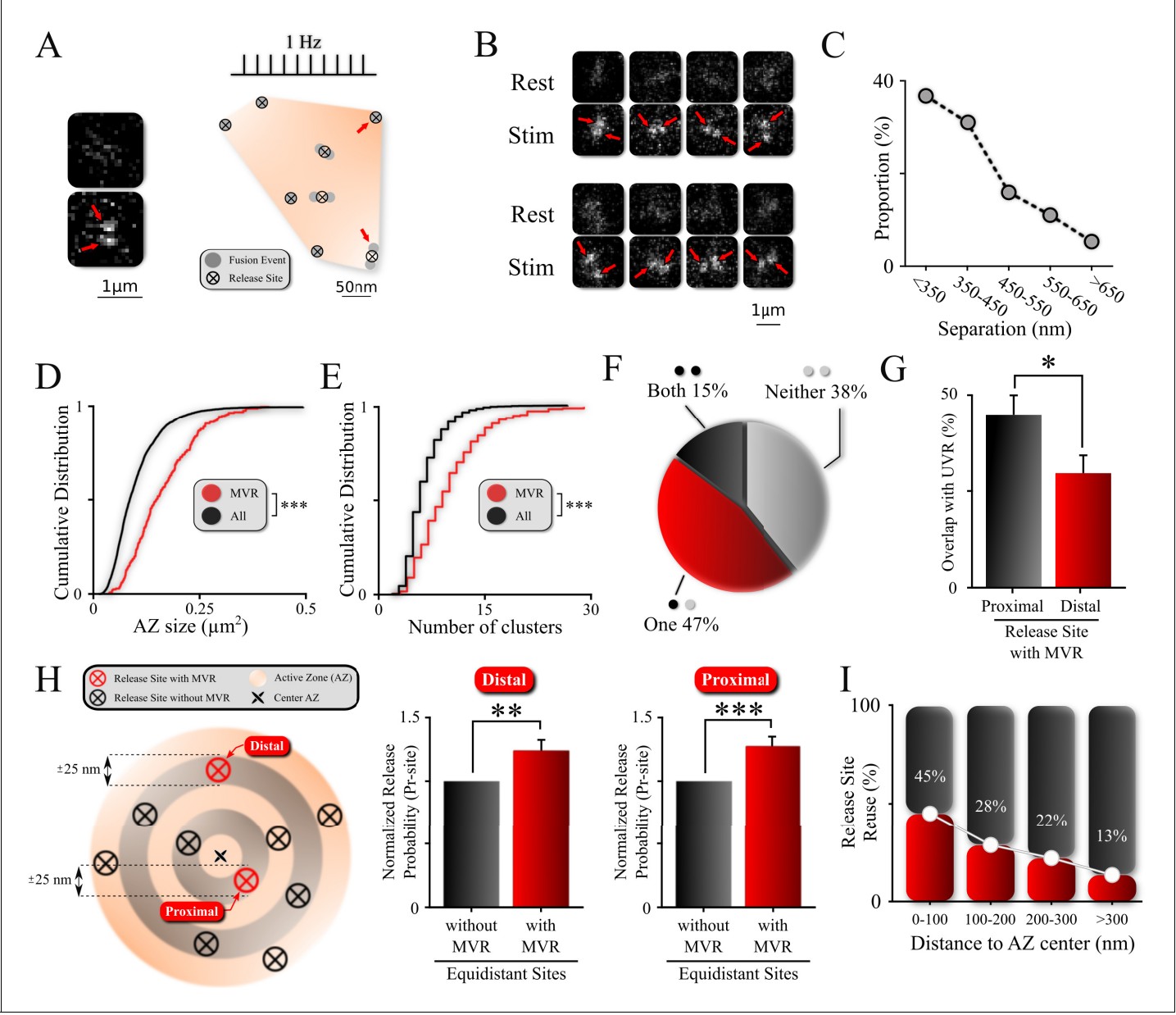

**Figure 1.** Non-uniform spatial features of MVR events and release sites within an AZ. (**A**) Sample spatial distribution of ten UVR events and a single MVR (arrows) event within a hippocampal bouton evoked by 120 APs at 1 Hz. Release sites were defined using a hierarchical clustering algorithm with a cluster diameter of 50 nm (***Maschi and Klyachko, 2017***) and are shown by crossed circles. Images (before and after 1 AP stimulation) show a sample MVR event highlighted by arrows. (**B**) Examples of MVR events in different boutons. (**C**) Proportion of MVR events as a function of intra-event separation distances. (**D, E**) Cumulative distributions of AZ area (**D**) and number of clusters/release sites (**E**) for all recorded boutons (black) and boutons exhibiting MVR events (red). (**F**) Spatial overlap of MVR and UVR events. Percentages of MVR events in which none, one or both events in the pair occurred at release sites that also harbored at least one UVR event. (**G**) Probability of reuse by UVR events of more proximal vs. more distal release sites engaged in MVR event pairs. (**H**) Analysis of release probability ($P_r$-site) of more distal (left bars) and more proximal (right bars) release sites engaged in MVR event pairs compared to other release sites equidistant to the AZ center within ±25 nm but not engaged in the MVR event during the observation period (shown schematically on the left). (**I**) Proportion of release sites that are reused at least once during the observation period as a function of the distance to the AZ center. Numbers shown represent average release site re-use in a given bin. N = 3781 (UVR) and 245 (MVR) events, from 90 dishes from 11 independent cultures; *p<0.05, **p<0.01, ***p<0.001. Two-sample KS-test (**D, E**); Chi-square test (**G**); Paired t-test (**H**).

The online version of this article includes the following figure supplement(s) for figure 1:

**Figure supplement 1.** Overlap of MVR and UVR events determined by proximity analysis.

in the same bouton is negligible, as we have previously used 3D FIB-SEM reconstruction of our neuronal cultures to show that the average bouton-to-bouton distance is an order of magnitude larger than the average event-to-event distance (*Maschi and Klyachko, 2017*). At the other end of the spectrum, it is important to note that this mixture-model fitting approach does not reliably fit the subset of double fusion events that occur so close as to be nearly overlapping. We examine this subset of unresolved MVR events using additional computational tools that are presented in the subsequent sections.

## MVR events preferentially occur at release sites with higher release probability

We first asked how the incidence of MVR is distributed in the synapse population. Previous studies suggest that the synaptic release probability is a strong predictor of a propensity for MVR (*Christie and Jahr, 2006*; *Huang et al., 2010*). Moreover, we know that AZ size is a major determinant of synaptic release probability (*Holderith et al., 2012*; *Matz et al., 2010*). We thus explored the relationship between AZ size (see 'Materials and methods' for details) and the probability of observing MVR events at individual synaptic boutons. We also used the number of release sites per bouton as another, functional measure of synaptic release probability. Individual release sites within each bouton were defined using a hierarchical clustering algorithm with a cluster diameter of 50 nm (*Maschi and Klyachko, 2017*). Boutons at which at least one MVR event was observed had a significantly larger AZ (*Figure 1D* and *Table 1*; N = 3781 (UVR) and 245 (MVR) events, 90 dishes from 11 independent cultures; p<0.001, two-sample KS-test) and a significantly larger number of release sites than the synapse population overall (*Figure 1E*; *Table 1*). These results suggest that MVR events are more likely to occur at boutons that have larger AZs and higher overall release probability.

In addition to detecting a variable propensity for MVR across the synapse population, we were interested in whether there is similar variability among release sites within the same synaptic bouton. In other words, are all release sites within a single bouton equally likely to support vesicle fusion involved in both UVR and MVR events or are there specific characteristics of individual release sites that make them more or less likely to support one type of fusion event over the other? We looked at the extent of release site overlap between MVR and UVR events, defining it as a 'full overlap' when *both* release sites involved in MVR were also observed as release sites during UVR events, 'partial overlap' when only one release site involved in MVR also served as a release site during UVR, and 'no overlap' when neither release site involved in an MVR event was observed as a release site for a UVR event during our observation period. We found full overlap with UVR for ~15% of MVR events, whereas ~47% of MVR events showed partial overlap, and ~38% showed no overlap (*Figure 1F*; *Table 1*). These results did not depend on the specific definition of release sites because we obtained essentially the same breakdown using proximity analysis of individual events without defining release sites using any clustering algorithms (*Figure 1—figure supplement 1*; *Table 1*).

This observation suggests that release sites can be involved in both UVR and MVR events, but that the likelihood that they are involved in one versus the other is not uniform (explored in more detail below). We also note that because we can only observe release events over a relatively short period (limited to 120 s by natural synapse displacement [*Maschi and Klyachko, 2017*]), our results cannot be interpreted as indicating that there are some specialized release sites that *only* support MVR or UVR. MVR is a relatively low probability event (~10% of release events are MVR under our experimental conditions), making much longer recordings necessary to determine whether sites are never involved in MVR; indeed, infinitely long recordings are required to answer this question definitively.

To understand better why some sites may be more likely to support MVR events than others, for each of the two release sites involved in an MVR event, we determined the probability that the same release site is also involved in one or more UVR events during our observation period. We found that this probability was location-dependent: of the two release sites involved in a given MVR event, the one more proximal to the AZ center was significantly more likely to be also involved in UVR events than the more distal site (*Figure 1G*; *Table 1*). Thus, the probability that a given release site was involved in both UVR and MVR events formed a gradient from AZ center to periphery, supporting the notion of heterogeneity of release site properties across a single AZ.

To further examine this notion, we quantified the release probability at each release site ($P_r$-site) based on the number of release events detected during the 120 s observation period. We then

**Table 1.** Data values and statistical analyses.

Columns represent (from left to right): figure/panel number; experimental conditions; number of samples (synapses, dishes and cultures); mean values and standard errors of the means (SEM); statistical test used for comparison; and the P-value resulting from the statistical comparison.

| Figure number | Conditions | NSyn | NDishes | NCultures | Mean ± SEM | Statistical test | Pval |
|---|---|---|---|---|---|---|---|
| *Figure 1D* | All | 3781 | 90 | 11 | 0.1014 ± 0.0009 | Two-sample KS-test | <0.001 |
| | MVR | 245 | 90 | 11 | 0.164 ± 0.005 | | |
| *Figure 1E* | All | 3781 | 90 | 11 | 6.57 ± 0.05 | Two-sample KS-test | <0.001 |
| | MVR | 245 | 90 | 11 | 9.8 ± 0.03 | | |
| *Figure 1G* | Proximal | 245 | 90 | 11 | 0.45 ± 0.05 | Chi-square test | 0.0386 |
| | Distal | 245 | 90 | 11 | 0.29 ± 0.05 | | |
| *Figure 1H* | Distal MVR/UVR | 66 | 90 | 11 | 1.24 ± 0.09 | Paired t-test | 0.006 |
| | Proximal MVR/UVR | 95 | 90 | 11 | 1.28 ± 0.08 | Paired t-test | <0.001 |
| *Figure 2C* | UVR | 151 | 90 | 11 | 0.32 ± 0.02 | Two-sample KS-test | <0.001 |
| | MVR | 144 | 90 | 11 | 0.47 ± 0.02 | | |
| *Figure 3B* | Proximal MVR/Proximal MVR | 245 | 90 | 11 | 1 ± 0.03 | Chi-square test | 0.017 |
| | Distal MVR/Proximal MVR | 245 | 90 | 11 | 0.65 ± 0.02 | | |
| *Figure 3C* | Larger MVR | 245 | 90 | 11 | 214 ± 7 | Two-sample t-test | 0.0058 |
| | Smaller MVR | 245 | 90 | 11 | 242 ± 7 | | |
| *Figure 3D* | MVR | 245 | 90 | 11 | $y = 3.108 + 0.02106\,x$ | Linear fit | <0.001 |
| *Figure 3E* | MVR <400 nm | 129 | 90 | 11 | 9.8 ± 0.7 | Two-sample t-test | <0.001 |
| | MVR >400 nm | 115 | 90 | 11 | 14.3 ± 0.9 | | |
| *Figure 4A* | EGTA MVR Linear fit | 225 | 57 | 11 | $y = 8.801 + 0.00556 \times$ | Linear fit | 0.264 |
| | MVR <400 nm | 156 | 57 | 11 | 10.1 ± 0.8 | Two-sample t-test | 0.131 |
| | MVR >400 nm | 69 | 57 | 11 | 12.3 ± 1.2 | | |
| *Figure 4B* | Ctrl MVR | 245 | 90 | 11 | 0.021 ± 0.005 | One-way analysis of covariance (ANOCOVA) | 0.022 |
| | EGTA MVR | 225 | 57 | 11 | 0.006 ± 0.005 | | |
| *Figure 4C* | Ctrl MVR | 245 | 90 | 11 | 52 ± 3 | Chi-square test | <0.001 |
| | EGTA MVR | 225 | 57 | 11 | 69 ± 3 | | |
| *Figure 4D* | Larger MVR | 225 | 57 | 11 | 178 ± 7 | Two-sample t-test | <0.001 |
| | Smaller MVR | 225 | 57 | 11 | 216 ± 7 | | |
| *Figure 4E* | Ctrl Pr = 0.042 | 2417 | 90 | 11 | 104 ± 8 | Two-sample t-test | <0.001 |
| | EGTA Pr = 0.042 | 2338 | 57 | 11 | 62 ± 5 | | |
| | Ctrl Pr = 0.033 | 2417 | 90 | 11 | 93 ± 3 | Two-sample t-test | <0.001 |
| | EGTA Pr = 0.033 | 2338 | 57 | 11 | 72 ± 2 | | |
| | Ctrl Pr = 0.025 | 2417 | 90 | 11 | 107 ± 2 | Two-sample t-test | <0.001 |
| | EGTA Pr = 0.025 | 2338 | 57 | 11 | 83 ± 1 | | |
| | Ctrl Pr = 0.017 | 2417 | 90 | 11 | 124 ± 1 | Two-sample t-test | <0.001 |
| | EGTA Pr = 0.017 | 2338 | 57 | 11 | 100 ± 1 | | |
| | Ctrl Pr = 0.008 | 2417 | 90 | 11 | 154.6 ± 0.7 | Two-sample t-test | <0.001 |
| | EGTA Pr = 0.008 | 2338 | 57 | 11 | 129.8 ± 0.8 | | |
| *Figure 4F* | Ctrl MVR | 2417 | 90 | 11 | $y = a + b * x^c$ | Fit | 0.0030 |
| | EGTA MVR | 2338 | 57 | 11 | a = 17.951 | | |
| | | | | | b = 1.0049e+09 | | |
| | | | | | c = 5.5575 | | |

*Table 1 continued on next page*

*Table 1 continued*

| Figure number | Conditions | NSyn | NDishes | NCultures | Mean ± SEM | Statistical test | Pval |
|---|---|---|---|---|---|---|---|
| *Figure 4G* | Distal MVR/UVR | 52 | 57 | 11 | 1.29 ± 0.09 | Paired t-test | 0.0020 |
| | Proximal MVR/UVR | 77 | 57 | 11 | 1.5 ± 0.1 | Paired t-test | <0.001 |
| *Figure 3— figure supplement 1A* | UVR | 136 | 90 | 11 | 11.5 ± 0.8 | Two-sample t-test | 0.1262 |
| | MVR,<400 nm | 129 | 90 | 11 | 9.8 ± 0.7 | | |
| | UVR | 136 | 90 | 11 | 11.5 ± 0.8 | Two-sample t-test | 0.0232 |
| | MVR,>400 nm | 115 | 90 | 11 | 14.3 ± 0.9 | | |
| *Figure 3— figure supplement 1B* | UVR | 665 | 90 | 11 | y = 35.088-0.0064674 x | Linear fit | 0.207 |
| | UVR 0–100 nm | 285 | 90 | 11 | 34.9±0.6 | Two-sample t-test | 0.1376 |
| | UVR 200–300 nm | 109 | 90 | 11 | 33.4 ± 0.9 | | |
| *Figure 3— figure supplement 1D* | Synaptic vesicle diameters | NSyn = 93 NVesic = 806 | – | 3 | y = 48.109+0.0047331x | Linear fit | 0.161 |
| *Figure 4— figure supplement 1A* | Larger Ctrl MVR | 245 | 90 | 11 | 214 ± 7 | Two-sample t-test | 0.0058 |
| | Smaller Ctrl MVR | 245 | 90 | 11 | 242 ± 7 | | |
| | Larger EGTA MVR | 225 | 57 | 11 | 178 ± 7 | Two-sample t-test | <0.001 |
| | Smaller EGTA MVR | 225 | 57 | 11 | 216 ± 7 | | |
| | Larger Ctrl MVR | 245 | 90 | 11 | 214 ± 7 | Two-sample t-test | <0.001 |
| | Larger EGTA MVR | 225 | 57 | 11 | 178 ± 7 | | |
| | Smaller Ctrl MVR | 245 | 90 | 11 | 242 ± 7 | Two-sample t-test | 0.0116 |
| | Smaller EGTA MVR | 225 | 57 | 11 | 216 ± 7 | | |
| *Figure 4— figure supplement 1B* | Ctrl Pr = 0.042 | 2417 | 90 | 11 | 104 ± 8 | Two-sample t-test | <0.001 |
| | EGTA Pr = 0.042 | 2338 | 57 | 11 | 62 ± 5 | | |
| | Ctrl Pr = 0.033 | 2417 | 90 | 11 | 93 ± 3 | Two-sample t-test | <0.001 |
| | EGTA Pr = 0.033 | 2338 | 57 | 11 | 72 ± 2 | | |
| | Ctrl Pr = 0.025 | 2417 | 90 | 11 | 107 ± 2 | Two-sample t-test | <0.001 |
| | EGTA Pr = 0.025 | 2338 | 57 | 11 | 83 ± 1 | | |
| | Ctrl Pr = 0.017 | 2417 | 90 | 11 | 124 ± 1 | Two-sample t-test | <0.001 |
| | EGTA Pr = 0.017 | 2338 | 57 | 11 | 100 ± 1 | | |
| | Ctrl Pr = 0.008 | 2417 | 90 | 11 | 154.6 ± 0.7 | Two-sample t-test | <0.001 |
| | EGTA Pr = 0.008 | 2338 | 57 | 11 | 129.8 ± 0.8 | | |

compared $P_r$-site separately for each of the two sites involved in the MVR event to other release sites (i.e., those involved only in UVR events) located equidistantly from the AZ center in the same bouton (i.e. within a ± 25 nm band, *Figure 1H*). We observed that both release sites engaged in an MVR event had a significantly higher release probability than other equidistant, non-MVR-involved release sites in the same bouton (*Figure 1H*; *Table 1*).

We also observed more general patterns among release sites within a given AZ, including those involved in UVR, MVR, or both. First, release sites were highly heterogeneous in terms of release probability, which varied ~5 fold among release sites within the same AZ during the observation time ($P_r$-site range [0.008–0.042]). Second, we observed a spatial gradient of $P_r$-site, which decayed with distance from the AZ center (*Figure 1I*).

Taken together, these findings provide evidence of a marked heterogeneity of release site properties within the individual AZs, characterized by a gradient of $P_r$-site from the AZ center to the periphery. These results further suggest that, in addition to the radial distribution of release probability, release sites that have a higher propensity for MVR are characterized by a higher release probability than other sites equidistant to the AZ center.

## The spatiotemporal features of resolved MVR events are generalizable to MVR events that cannot be resolved

The well-separated MVR events analyzed above had sufficient spatial separation to allow each event in the pair to be individually localized (resolved events). However, because the AZ is very small over-all, a significant proportion of MVR events involve vesicle release at sites within a sub-diffraction distance from one another. Such unresolved events would not have been captured in our analyses thus far. Therefore, we next asked to what extent the findings relating to resolvable MVR events could be generalized to unresolved MVR events.

To identify unresolved MVR events, we took advantage of the quantal nature of vesicular release to distinguish MVR from UVR events based on amplitude (*Balaji and Ryan, 2007*; *Leitz and Kavalali, 2011*). At 2 mM extracellular $Ca^{+2}$ and at 37°C, over 90% of the events in hippocampal neurons are UVR (*Leitz and Kavalali, 2011*; *Maschi and Klyachko, 2017*). Thus, we analyzed individually each synaptic bouton with a minimum of five fusion events to determine the mean and intrasynaptic variability (standard deviation) of quantal event amplitude. We then set the threshold for MVR event detection at two standard deviations above the mean quantal event amplitude for that bouton (*Figure 2A,B*). On the basis of this analysis, we estimated that in our studies, MVR events represent ~9% of all release events. Of these events, we were able to robustly identify ~50% as MVR based solely on their amplitude (*Figure 2B*). This approach does not rely on spatial information, and thus is complementary to the mixture-model fitting approach that we used above; the MVR event populations that were identified by these two approaches are partially overlapping (~20%; data not shown).

The identified MVR events were then analyzed on the basis of asymmetry considerations, using an asymmetric Gaussian model fit that takes into consideration the pixelated nature of the image to determine the width (sigma) of the Gaussian fit in the maximal (longitudinal, $\delta_1$) direction and the minimal (transverse, $\delta_2$) direction. The ratio $\delta_1/\delta_2$–1 (asymmetry score) represents an estimate of asymmetry of the double-event image, which correlates with the distance between the two sub-diffraction events forming the image (*DeCenzo et al., 2010*). Distributions of asymmetry scores for the single and double events indicate that they represent two distinct populations (*Figure 2C*; *Table 1*) and thus validate our approach to robustly distinguish unresolved MVR events from UVR events.

We then examined the spatiotemporal features of the unresolved MVR events. First, we observed that unresolved MVR events preferentially have smaller asymmetry scores (*Figure 2D*) and thus tend to occur at smaller separation distances, similarly to the resolved MVR events (*Figure 1C*). Next, we examined the localization of unresolved MVR events relative to the AZ center/periphery. We observed that more asymmetrical (more spatially separated) events occurred closer to the AZ center, whereas symmetrical events tend to be more peripheral (*Figure 2E*, top). An equivalent calculation for the resolved MVR events (see figure legend for details) showed a very similar relationship (*Figure 2E*, bottom), supporting the notion that the two subpopulations of MVR events have similar spatial properties. Finally, as described for the resolved MVR events above, we examined the extent to which release sites were used for either MVR or UVR, or both. Only a subpopulation of strongly overlapping MVR events (asymmetry score <0.5) were used in this analysis because these highly symmetrical events could be well-approximated by a single symmetrical Gaussian fit, making this analysis comparable to that of the resolved MVR events. The extent of overlap of MVR and UVR events at individual release sites was comparable between unresolved and resolved MVR events (*Figures 2F* and *1F*), with overlap more likely occuring closer to the AZ center in both cases (*Figures 2F* and *1G*). These data suggest that unresolved and resolved MVR events have comparable spatial features. Thus, our observations can likely be generalized across the entire population of MVR events.

## Temporal desynchronization of release events comprising MVR

We next asked whether there is also heterogeneity within the temporal properties of MVR events. Previous studies present evidence of temporal 'jitter' on a millisecond timescale (~1–5 ms) within the release pair comprising an MVR event at both excitatory and inhibitory cerebellar synapses (*Auger et al., 1998*; *Auger and Marty, 2000*; *Crowley et al., 2007*; *Malagon et al., 2016*; *Rudolph et al., 2011*). Upon initial inspection of resolved MVR events, we noticed that one of the two fusion events comprising an MVR is often noticeably larger in amplitude than the other (*Figure 3A*). In our experimental approach, action potentials are synchronized with the beginning of

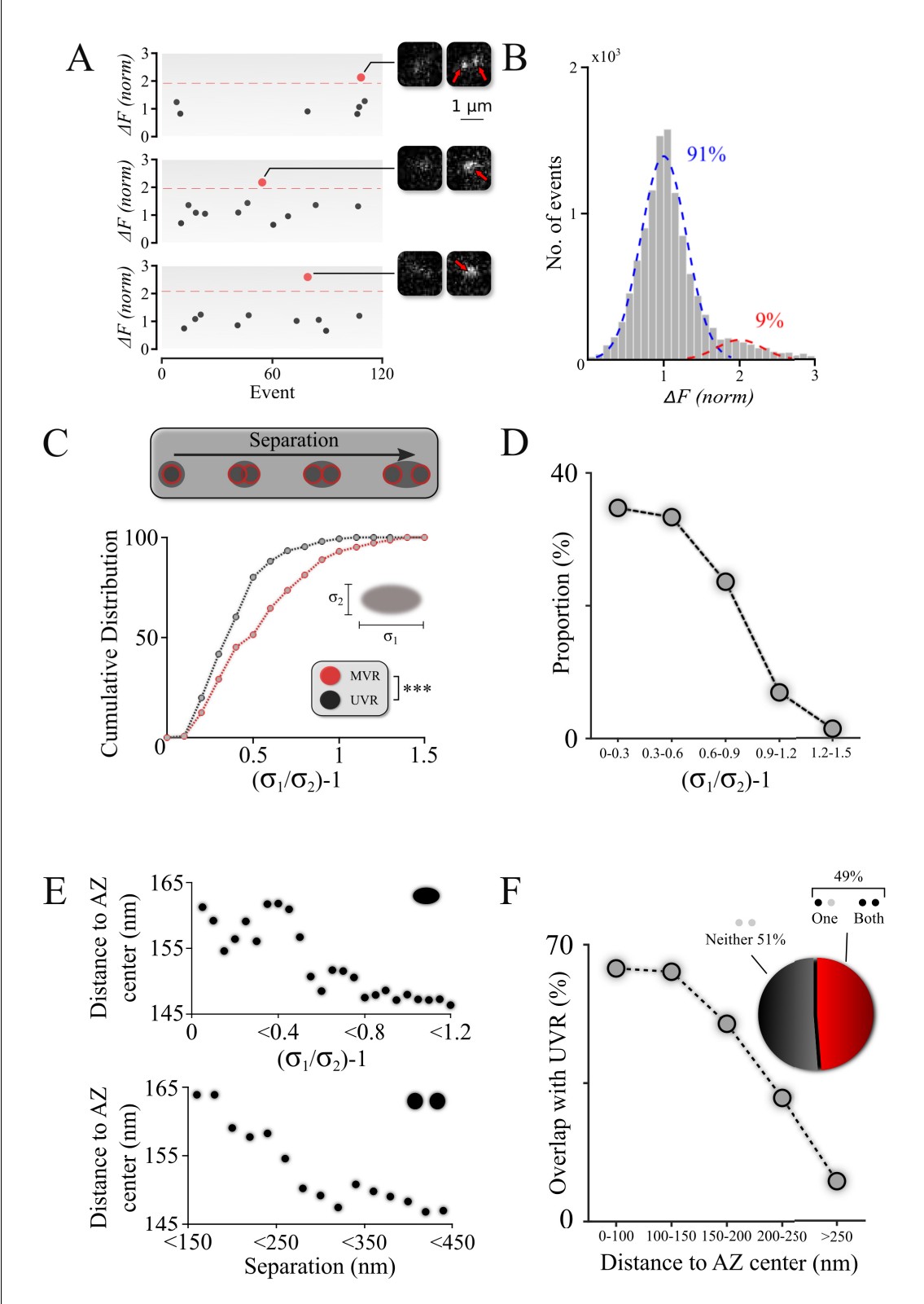

**Figure 2.** Spatiotemporal features of resolved MVR events generalize to unresolved MVR events. (**A**) MVR (red) and UVR (black) events were separated on the basis of the event amplitude. Examples of an identified resolved MVR event (top) and two unresolved MVR events (middle and bottom) are shown with corresponding images. (**B**) An intensity histogram for all detected events from panel (**A**) reflects the quantal nature of fusion events. Gaussian fits to the first peak (UVR events, blue) and second peak (MVR events, red) and their relative abundances are shown. (**C**) Asymmetry analysis of

*Figure 2 continued on next page*

*Figure 2 continued*

unresolved MVR events vs UVR events. The asymmetry score was calculated using asymmetrical Gaussian fit to the event image to determine maximal ($\delta_1$) and minimal ($\delta_2$) width (*insert*). (D) Proportion of unresolved MVR events as a function of asymmetry score, which correlates with intra-event separation distances. (E) Mean distance to the AZ center for unresolved (*top*) and resolved (*bottom*) MVR events. Distance was calculated from the peak of the Gaussian fit for unresolved MVR events and from the center of the line connecting two fusions within the resolved MVR events. (F) Probability of overlap of unresolved MVR and UVR events in the same bouton as a function of distance to the AZ center. Event overlap was determined using proximity analysis with a radius of 25 nm. Only more symmetrical MVR events (asymmetry score <0.5) were included in this analysis. Points represent the proportion of MVR events within a given distance band that overlapped with UVR events. *Pie chart:* proportion of unresolved MVR events that overlap or not with UVR events during the observation period. N = 151 (UVR) and 144 (MVR) events, 90 dishes from 11 independent cultures. *** p<0.001; Two-sample KS-test (C).

the frame acquisition; thus, considering the quantal nature of fusion events, we hypothesized that this difference in amplitude reflects imperfect synchronization between the two release events (if one release event occurs later in the recording frame, we would expect to collect a smaller number of photons for this delayed event, i.e., smaller amplitude) (*Figure 3A*, top).

To test this hypothesis, we looked at the relationship between the relative amplitude and the spatial organization of the release pairs comprising each MVR event. We observed that the larger amplitude event (assumed to occur first) was more likely to occur closer to the AZ center than the smaller one in the pair (*Figure 3B*; *Table 1*). Accordingly, the average distance to the AZ center was significantly shorter for the larger amplitude event relative to the event with the smaller amplitude within the pair (*Figure 3C*; *Table 1*). Most importantly, this amplitude difference was correlated with the distance between the two events within the pair (*Figure 3D,E*), such that larger amplitude differences were associated with larger separation distances (*Figure 3D,E*). This spatial organization parallels the gradient of release site release probability from the AZ center to the periphery (*Figure 1I*, and see below). We note that a component of the amplitude differences can arise from the uncertainty in determining the fusion event amplitude; we estimated this uncertainty to be ~10% (*Figure 3—figure supplement 1A*). Thus, the uncertainty in our measurements may account for the amplitude differences that we measured for the most closely spaced MVR events, but it does not account for the differences that we measured for MVR events that are further apart (*Figure 3—figure supplement 1A*). Moreover, the positive correlation between the amplitude difference and the spatial separation of the two events comprising MVR cannot be explained by random noise or by measurement uncertainty. We thus interpret our results as indicating that the amplitude difference between the two fusion events that comprise an MVR, at least in part, reflects imperfect synchronization.

Given the observed amplitude differences within the release pairs comprising an MVR event, and the acquisition duration of 40 ms per frame, we estimated the maximal time difference between the two events comprising MVR to be less than ~4 ms for the majority of MVR events in our recordings. We note that this value overestimates the true extent of desynchronization because, as noted above, a component of the amplitude difference arises from uncertainty in the amplitude measurement itself. Thus, we estimate the maximal time delay to be ~2 ms if the measurement uncertainty is factored in. These values are in a close agreement with previous reports, which found desynchronization within individual MVR events to be in the range of 1–5 ms (*Auger et al., 1998*; *Auger and Marty, 2000*; *Crowley et al., 2007*; *Malagon et al., 2016*; *Rudolph et al., 2011*).

We also considered the possibility that factors other than desynchronization, such as differences in vesicle size or cleft pH along the AZ plane, contribute to the difference in amplitude within MVR events and its spatial arrangement relative to the AZ center. We used Large-Area Scanning Electron Microscopy (LaSEM) micrographs of our cultures (*Maschi et al., 2018*) to determine the relationship between the size of vesicles positioned near the AZ (within 100 nm, defined previously as tethered vesicles; *Maschi et al., 2018*) and their position relative to the AZ center. Vesicle diameter did not exhibit any measurable change as a function of distance from the AZ center (*Figure 3—figure supplement 1C,D*), indicating that the amplitude differences that we observed are not due to systematic differences in vesicle size. We next examined cleft pH at different locales within the AZ. The peak amplitude of vGlut1-pHluorin signal during individual fusion events is determined in large part by the cleft pH. Thus, we measured peak pHluorin signal as a function of distance to AZ center. The vGlut1-pHluorin signal amplitude (*Figure 3—figure supplement 1B*) did not measurably change as

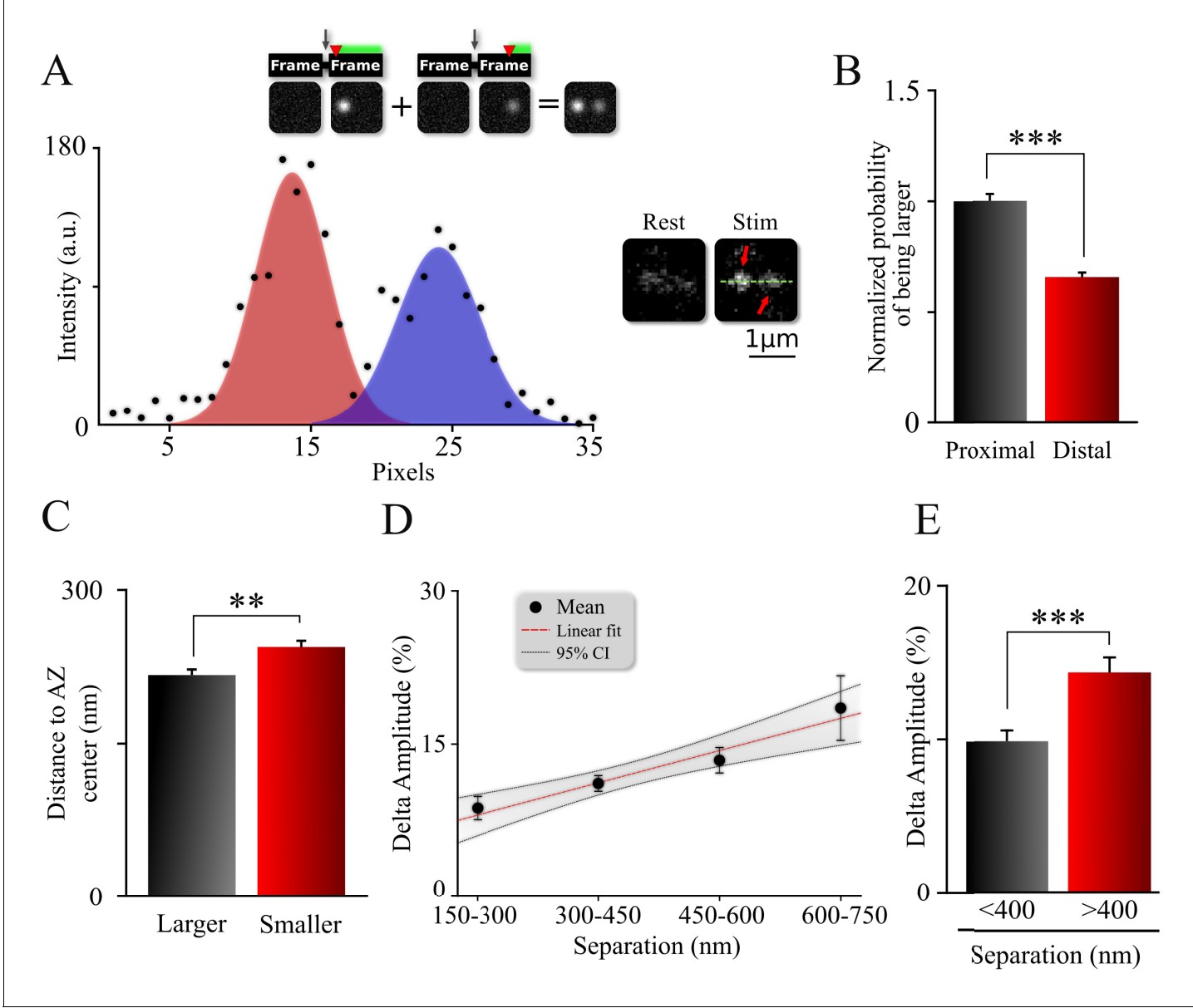

**Figure 3.** Spatiotemporal organization of release events comprising an MVR. (**A**) Sample image (*right*) and intensity profile (*left*) of an MVR event with noticeable difference in intra-event amplitude. The top insert shows a cartoon representation of a relationship between a time delay (*red arrow*) of the second fusion after an action potential and the resulting amplitude difference within an MVR event. (**B**) Probability that the proximal or distal event within MVR pairs is larger, normalized to that of the proximal event. (**C**) Distance to the AZ center from the larger and smaller events within MVR pairs. (**D, E**) Amplitude difference of the two events comprising MVR as a function of intra-event separation. Linear fit (**D**) and t-test of pooled data (**E**) are shown. *p<0.05, ***p<0.001; Chi-square test (**B**); Paired t-test (**C**); Two-sample t-test (**E**). N = 245 MVR events, from 90 dishes from 11 independent cultures.

The online version of this article includes the following figure supplement(s) for figure 3:

**Figure supplement 1.** Amplitude difference within the MVR event pairs is not due to measurement uncertainty, changes in vesicle size or cleft pH within the AZs.

**Figure supplement 2.** Double events do not result from asynchronous release overlapping temporally with synchronous events.

a function of distance from the AZ center, suggesting that a gradient of cleft pH is unlikely to explain the differences in MVR event amplitude. Finally, we note that in our imaging experiments, dozens of synapses are positioned in random orientation and are recorded simultaneously. Thus, in any given recording, some boutons have the center of the AZ in focus, whereas in others, only the periphery is

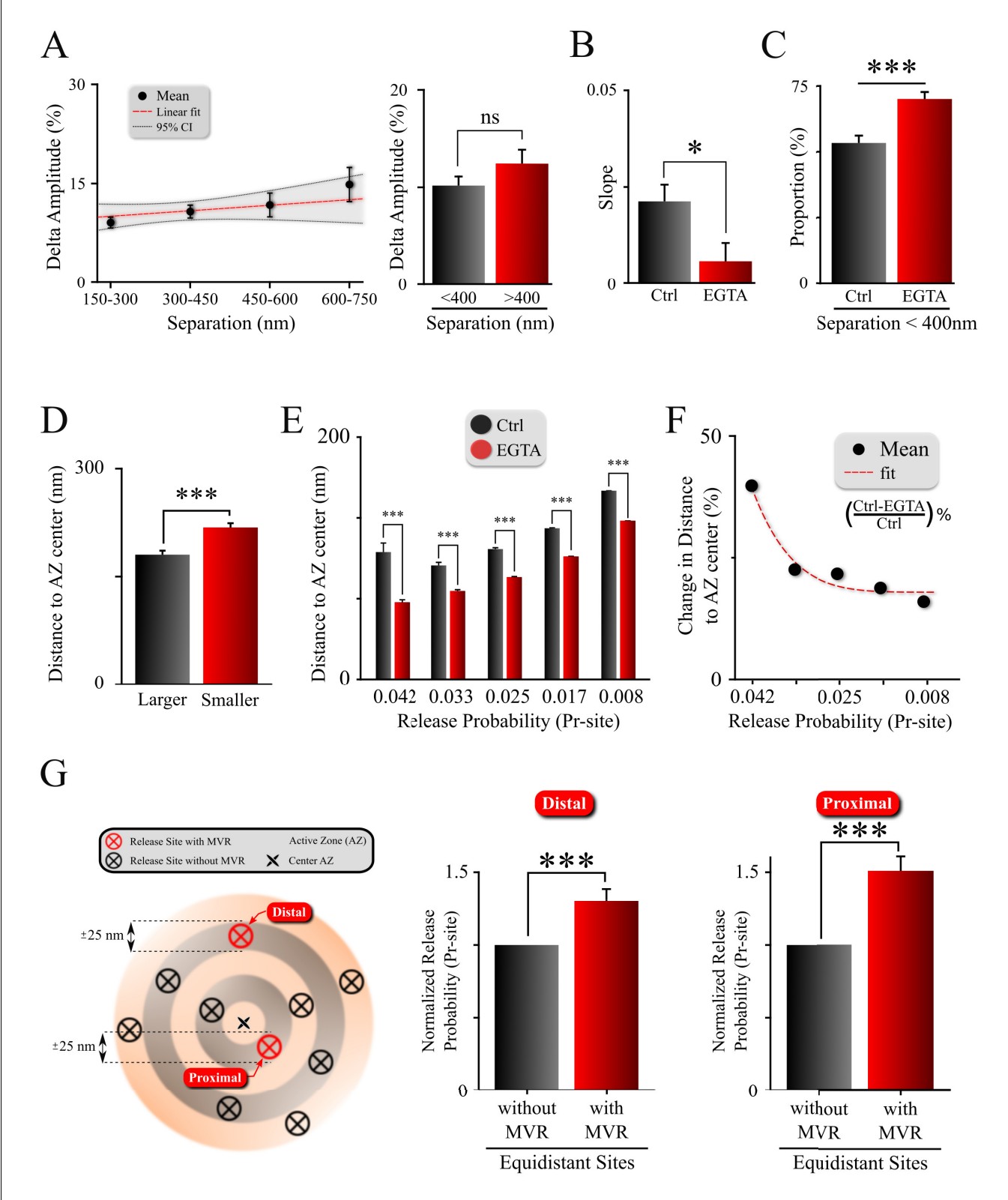

**Figure 4.** Spatiotemporal features of MVR events and release site properties are calcium-dependent. (**A**) Effect of EGTA on the correlation between the spatial separation and amplitude difference between two events comprising an MVR. (**B**) The effect of EGTA was assessed by comparing the slopes of the correlation in panel (**A**) in control (from *Figure 3D*) and EGTA (from **A**) conditions. (**C**) Proportion of MVR events with intra-event separation <400 nm in EGTA and control conditions. (**D**) Distance to the AZ center from the larger and smaller events within MVR pairs in the presence of EGTA. (**E,F**)
*Figure 4 continued on next page*

*Figure 4 continued*

Average distance to the AZ center (E) and its relative change (F) for individual release sites binned on the basis of their release probability, in EGTA and control conditions. (G) Release probability of more distal (left bars) and more central (right bars) release sites engaged in MVR event pairs compared to all other release sites equidistant to the AZ center within ±25 nm, in the presence of EGTA. ***p<0.001; ns, not significant. Statistical tests were as follows: two-sample t-test (A,E); one-way analysis of covariance (ANOCOVA) (B); Chi-square test (C); and Paired t-test (D,G). Control: N = 245 MVR events from 90 dishes from 11 independent cultures. EGTA: N = 225 MVR events from 57 dishes from 11 independent cultures.

The online version of this article includes the following figure supplement(s) for figure 4:

**Figure supplement 1.** Violin plots of the amplitude differences within MVR event pairs and the effects of EGTA.

in focus, and so on for all possible configurations in-between. Thus, there is no bias towards any one configuration or tilt of the AZ relative to the imaging plane that can lead to systematically larger fusion event amplitude at the AZ center versus that in the periphery. Indeed, this notion is highlighted by the fact that, in our measurements, the average release event amplitude is indistinguishable between events located at the AZ center versus those in the periphery (*Figure 3—figure supplement 1B*, *insert*). Thus, we conclude that the observed differences in amplitude within the release pair comprising an MVR event are unlikely to result from a bias in AZ position relative to the imaging plane.

In summary, our results support the notion that vesicle release associated with MVR is desynchronized and follows a specific spatial organization with respect to the center of the AZ. Specifically, the first of the two events in the MVR pair is preferentially located closer to the AZ center. This spatial organization of MVR events parallels our finding that release probability and the probability that a release site is involved in an MVR also follows a center-to-periphery spatial gradient.

## Double events do not result from asynchronous release overlapping temporally with synchronous events

Given the intrinsically limited temporal resolution of our imaging tools, we must consider the possibility that the double events that we observed arose not from MVR, but rather from asynchronous release events generated by preceding stimulation and temporarily overlapping with UVR. We addressed this possibility in five complementary ways.

First, if the double events do in fact arise from overlap of synchronous release with asynchronous events generated by preceding stimulation, the probability of observing such double events should increase over the course of the stimulus train. However, we detected no increase in the double event probability during the stimulus train (*Figure 3—figure supplement 2A*, *Table 1*). In fact, the probability of observing double events is slightly higher for the first stimulus, for which there is no preceding stimulation. These results argue against any significant contribution of asynchronous release to the observed double-release events.

Second, synchronous release events, including those associated with MVR, are time-locked with the stimulus and are acquired for the entire duration of the frame. Thus, the amplitude distribution of MVR events should appear as a single Gaussian peak centered at ~2 q value (twice the amplitude of a UVR event), as indeed is the case in our measurements (*Figure 2B*). By contrast, asynchronous release is, by definition, not time-locked with the stimulus and thus occurs randomly at any time during the acquisition frame. As a result, single asynchronous release events are acquired for a wide range of durations that are less than a full frame, and thus must have a skewed non-Gaussian amplitude distribution shifted towards smaller values than those produced by a full-size synchronous event. We confirmed that this is the case using asynchronous events that we recorded in 3 mM $Sr^{2+}$ in otherwise identical conditions (*Figure 3—figure supplement 2B*, *Table 1*). Thus, we conclude that the double events in our recordings are synchronous events and have properties that are distinct from those of asynchronous release.

Third, we compared the spatiotemporal properties of the double events in our recordings with that of asynchronous release events recorded in 3 mM $Sr^{2+}$. The asynchronous release events detected in the same frame in the same bouton (*Figure 3—figure supplement 2C*) did not exhibit the spatiotemporal patterns evident for the double events (*Figure 3D*). The fact that the spatiotemporal properties of the double events are distinct from those of asynchronous events supports our conclusion that the double events in fact reflect instances of MVR.

Fourth, the synchronicity of double events in our recordings is relatively high, arguing against a significant contribution from asynchronous release. As mentioned above, we estimate that the double events in our recordings are synchronous with the stimulation within a few milliseconds. This result is consistent with previous reports of MVR, and aligns with the accepted definition of MVR events (*Auger et al., 1998*; *Auger and Marty, 2000*; *Crowley et al., 2007*; *Malagon et al., 2016*; *Rudolph et al., 2011*).

Finally, we note that previous studies find minimal asynchronous release evoked by 1 Hz stimulation under nearly identical experimental conditions (37˚C, 2 mM extracellular Ca$^{2+}$) (*Raingo et al., 2012*). Thus, at least five complimentary lines of evidence strongly suggest that double events in our recordings do indeed reflect synchronous MVR events, with no significant contribution of asynchronous release.

## Spatiotemporal features of MVR events and release site properties are calcium-dependent

We next set out to explore the mechanistic origin of the spatial organization of MVR events relative to the AZ center. Our previous study suggested a possible role of calcium in the spatial regulation of release site properties because we observed that release site usage is regulated in an activity-dependent manner, such that site usage shifts towards the periphery during trains of activity (*Maschi and Klyachko, 2017*). Previous studies have also found that the propensity for MVR events increases as extracellular calcium levels increase (*Leitz and Kavalali, 2011*). However, whether calcium regulates the spatiotemporal organization of MVR events and/or the properties of the release sites remains unknown. To test this possibility, we pre-incubated neurons with a cell-permeable calcium chelator EGTA-AM (30 μm) for 20 min. EGTA does not directly affect vesicle release because it is too slow to buffer rapid calcium rise near voltage-gated calcium channels (VGCCs), but it is effective in buffering the ensuing slower calcium elevation caused by diffusion. We observed several effects of EGTA on the spatiotemporal distribution of MVR events. First, EGTA affected MVR event desynchronization: although the amplitudes of the two fusion events comprising MVR were still different, the difference no longer depended on their separation distance (*Figure 4A,B*; *Table 1*; N = 225 synapses, from 57 dishes from 11 independent cultures). Second, in the presence of EGTA, a larger proportion of MVR events occurred at shorter intra-event distances (*Figure 4C*; *Table 1*), that is, pairs of release events comprising MVR were more likely to be closer to each other in the presence of EGTA. In addition, the average distances from both events in the MVR pair to the AZ center were significantly shortened in the presence of EGTA (*Figure 4—figure supplement 1A*, *Table 1*). Thus, calcium buffering causes MVR events to occur at shorter separation distances and more proximal to the AZ center. However, the preferential localization of the larger (earlier) event in the MVR pair closer to the AZ center was still observed in the presence of EGTA (*Figure 4D*, *Table 1*). These effects of buffering calcium diffusion with EGTA suggest that several major spatiotemporal features of MVR events are determined, in part, by calcium diffusion following an action potential.

In line with the idea that the spatiotemporal features of MVR events reflect release site heterogeneity, we observed that the preferential utilization of more central release sites was exacerbated in the presence of EGTA (*Figure 4E*, *Figure 4—figure supplement 1B*; *Table 1*). Interestingly, buffering intraterminal calcium had a larger effect on more central release sites with higher P$_r$-site values than on the more peripheral sites with lower P$_r$-site (*Figure 4F*). These results are in line with our earlier findings that the release site utilization shifts in the opposite direction (i.e. from AZ center towards periphery) when intraterminal calcium is elevated during high-frequency stimulation (*Maschi and Klyachko, 2017*). It is also consistent with the shorter separation distance within MVR events in the presence of EGTA (*Figure 4C*), and the shorter distances of MVR events to the AZ center in the presence of EGTA (*Figure 4—figure supplement 1A*). Indeed, the preferential use of sites with higher release probability (as compared to other, equidistant sites) during MVR events persists in the presence of EGTA (*Figure 4G*; *Table 1*), while the utilization of these sites shifts closer to the AZ center in the presence of EGTA.

Together, these observations suggest that the gradient of release site properties, as well as the spatiotemporal features of MVR events, are, in part, determined by the calcium concentration landscape across the AZ.

## Discussion

Although MVR is well established as a ubiquitous release mechanism that occurs at many types of synapses (*Rudolph et al., 2015*), its spatiotemporal organization and regulation within the AZ is largely unknown. We took advantage of our ability to detect individual release events at central synapses with nanoscale precision to reveal three major findings: (i) release sites within the same AZ have highly heterogeneous properties, reflecting a gradient of release probability that decreases from the AZ center to periphery; (ii) MVR events exhibit non-uniform patterns of spatial and temporal organization that parallel the center-to-periphery organization of release site release probability; and (iii) both the gradient of release site properties and the spatiotemporal features of MVR are determined, in part, by the intraterminal calcium elevation following an action potential. Together, these results suggest that the non-uniform spatiotemporal dynamics of MVR events arise from the heterogeneity of release site properties within the individual AZs.

Our results suggest a model of MVR events in which the earlier of the two events in the MVR pair is similar to a UVR event, in that it occurs closer to the AZ center because release sites with higher release probability are localized preferentially more proximally to the AZ center. A second release event is then triggered occasionally with a short delay after the same action potential, at a more peripheral release site that is primed for release, in part due to calcium spread from the AZ center. This notion is supported by the effects of calcium buffering with EGTA and by our observations that both events in the MVR pair occur at sites with higher release probability than other equidistant sites in the same bouton. These observations reveal a previously unrecognized complex landscape of release probability within the AZ, which determines the spatial organization of MVR, and arises, in part, from a calcium concentration gradient across individual AZs.

What are the molecular underpinnings of release site heterogeneity? Recent nanoscopy studies indicate that release sites colocalize with nanoclusters of presynaptic docking factors, such as RIM1/2 (*Tang et al., 2016*), which have been suggested to control the recruitment and clustering of VGCCs at the AZ via the RIM binding protein RIM-BP (*Davydova et al., 2014*; *Hibino et al., 2002*; *Wang et al., 2000*). RIM1/2 nanoclusters are more likely to be located near the center of the AZ than in the periphery, suggesting a possible structural basis for the gradient of release site properties that we observed in this work (*Tang et al., 2016*). Moreover, the enrichment of many scaffold/docking proteins, including RIM, Bassoon, and Munc13, within their clusters is dynamic (*Bademosi et al., 2016*; *Glebov et al., 2017*; *Smyth et al., 2013*; *Tang et al., 2016*; *Weyhersmüller et al., 2011*). Thus, the heterogeneity of release site properties may arise, in part, from variability in cluster architecture, such as cluster size or the relative enrichment of tethering/docking/priming factors.

An additional source of heterogeneity could be the fact that variable fractions of many critical components of release machinery, including VGCCs, Syntaxin-1, and Munc18, are mobile within the AZ (*Glebov et al., 2017*; *Schneider et al., 2015*; *Smyth et al., 2013*). For example, a large proportion (>50%) of VGCCs are mobile in the AZ plane and this mobility is calcium-dependent (*Schneider et al., 2015*). How VGCC mobility is spatially controlled has not been explored, but differential VGCC stability at the AZ center versus the periphery could account, in part, for the heterogeneity of the calcium rise across the bouton. VGCC mobility could also affect coupling between the channels and the vesicles (*Eggermann et al., 2012*; *Miki et al., 2017*). This could explain the differential effect of EGTA on peripheral versus more central release sites.

Another possibility is that release site properties are determined, in part, by extrinsic factors. For example, release site refilling and vesicle retention at release sites depends on actin and myosins (*Maschi et al., 2018*; *Miki et al., 2016*). Thus, a non-homogeneous spatial distribution of actin cytoskeleton could contribute to differential release site usage across a bouton. In addition, calcium influx at least through some subtypes of VGCCs is also modulated by the balance of phosphorylation/dephosphorylation by CDK5 and calcineurin (*Kim and Ryan, 2013*), and is directly regulated by a number of presynaptic proteins, such as ELKS (*Liu et al., 2014*) and Munc13 (*Calloway et al., 2015*). Although the precise factors that drive release site heterogeneity remain to be elucidated, our results reveal a previously unknown level of structural and functional organization among vesicle release sites within individual AZs, with specific implications for the spatiotemporal dynamics of MVR events.

## Materials and methods

### Neuronal cell cultures

Neuronal cultures were produced from rat hippocampus as previously described (*Peng et al., 2012*). Briefly, hippocampi were dissected from E16 pups, dissociated by papain digestion, and plated on coated glass coverslips. Neurons were cultured in Neurobasal media supplemented with B27. All animal procedures conformed to the guidelines approved by the Washington University Animal Studies Committee.

### Experimental design

All live imaging measurements were replicated in more than 100 boutons derived from 57 to 90 coverslips from 11 independent cultures (see *Table 1* for individual experiments). Most experiments were carried out in an unblended manner and no specific randomization strategy was used. Statistical computations were not performed to determine the optimal sample size for experiments.

### Lentiviral infection

VGlut1-pHluorin was generously provided by Drs Robert Edwards and Susan Voglmaier (University of California San Francisco) (*Voglmaier et al., 2006*). Lentiviral vectors were generated by the Viral Vectors Core at Washington University. Hippocampal neuronal cultures were infected at DIV3 as previously described (*Maschi and Klyachko, 2017*).

### Fluorescence microscopy

All experiments were conducted at 37°C within a whole-microscope incubator (In Vivo Scientific) at DIV16–19. Neurons were perfused with bath solution (125 mM NaCl, 2.5 mM KCl, 2 mM $CaCl_2$, 1 mM $MgCl_2$, 10 mM HEPES, 15 mM glucose, 50 µM DL-AP5, 10 µM CNQX adjusted to pH 7.4). Asynchronous release events were recorded using the same solutions, except that 3 mM $Sr^{2+}$ and 0 mM $CaCl_2$ were used in the bath. Fluorescence was excited with a Lambda XL lamp (Sutter Instrument) through a 100 × 1.45 NA oil-immersion objective and captured with a cooled CMOS camera (Hamamatsu). With this configuration, the effective pixel size was 65 nm. The focal plane was continuously monitored, and focal drift was automatically adjusted with ~10 nm accuracy by an automated feedback focus control system (Ludl Electronics). Field stimulation was performed by using a pair of platinum electrodes and controlled by the software via Master-9 stimulus generator (AMPI). Images were acquired using an acquisition time of 40 ms, one 45 ms before stimulation and one coincidently with stimulation (0 ms delay).

### Large-area scanning electron microscopy (LaSEM)

The LASEM methods and data used were published previously (*Maschi et al., 2018*). Briefly, cells were grown on 12 mm glass coverslips, were aspirated of media and were fixed in a solution containing 2.5% glutaraldehyde and 2% paraformaldehyde in 0.15 M cacodylate buffer with 2 mM $CaCl_2$ (pH 7.4) that had been warmed to 37 °C for one hour. The samples were then stained according the methods described by *Deerinck et al. (2010)*. Large areas (~330 × 330 µm) were then imaged at high resolution in a FE-SEM (Zeiss Merlin, Oberkochen, Germany) using the ATLAS (Fibics, Ottowa, Canada) scan engine to tile large regions of interest. High-resolution tiles were captured at 16,384 × 16,384 pixels at 5 nm/pixel with a 5 µs dwell time and line average of 2. The SEM was operated at 8 KeV and 900 pA using the solid-state backscatter detector. Tiles were aligned and exported using ATLAS 5.

### Image and data analysis

#### Event detection and localization

The fusion event localization at subpixel resolution was performed as previously described (*Maschi and Klyachko, 2017*) using Matlab and the uTrack software package, which was kindly made available by Dr Gaudenz Danuser's lab (*Aguet et al., 2013*; *Jaqaman et al., 2008*). The input parameters for the PSF were determined using stationary green fluorescent 40 nm beads.

Localization precision was determined directly from the least-squares Gaussian fits of individual events as described in *Thomann et al. (2002)* and *Thomann et al. (2003)* using in-built functions in

uTrack software (*Aguet et al., 2013*; *Jaqaman et al., 2008*). Spatial constraints of the vesicle lumen imply that only a few VGlut1-pHluorin molecules can be located within individual vesicles. Given our observations that the fluorescence signal evoked by vesicle fusion did not disperse significantly during our acquisition time of 40 ms, the small number of VGlut1-pHluorin molecules per vesicle and their lateral movement upon fusion, if present, do not strongly affect localization precision at the time at which our measurements are made.

Localization of resolved MVR events (*Figures 1*, *3* and *4*) was performed using a mixture-model multi-Gaussian fit as described in *Thomann et al. (2002)* and *Thomann et al. (2003)* using in-built functions in uTrack software (*Aguet et al., 2013*; *Jaqaman et al., 2008*).

Unresolved MVR events (*Figure 2*) were identified on the basis of the event amplitude. The single event amplitude and its variability were determined for each bouton individually. Photobleaching was accounted for by fitting the event intensity changes over time. The threshold for MVR event detection was set at two standard deviations above the mean single event amplitude determined individually for each bouton. Localization of unresolved MVR events was determined using an asymmetrical Gaussian model fit that was based on the minimization of the residuals.

## Definition of release sites
Release sites were defined using a hierarchical clustering algorithm based on built-in functions in Matlab as described previously (*Maschi and Klyachko, 2017*; *Maschi et al., 2018*; *Wang et al., 2016*). We have previously compared the results of this clustering analysis, obtained with the experimentally observed distribution of fusion events, with the the same number of simulated events distributed randomly across the same AZs (*Maschi and Klyachko, 2017*). We found that randomly distributed release events result in a very different pattern of clustering than the experimentally observed events, and do not reproduce the observed features of real release event clusters. The observed clusters thus do not arise from a random distribution of release events, but rather represent a set of defined and repeatedly reused release sites within the AZs.

## Release site release probability
The release probability of individual release sites was calculated based on the number of release events detected per release site, divided by the duration of the observation period. UVR and MVR events were counted equivalently in this analysis, with each of the two release events comprising an MVR counted independently as a single release event at the two release sites that harbored them.

## AZ dimensions and center
The AZ size was approximated on the basis of the convex hull encompassing all vesicle fusion events in a given bouton. This measurement is in a close agreement with the ultrastructural measurements of AZ dimensions (*Maschi and Klyachko, 2017*). AZ center was defined as the mean position of all fusion events in a given bouton.

## Event proximity analysis
To determine the spatial overlap of MVR and UVR events, a proximity analysis was performed in which overlap was defined as having at least one UVR event occurring within 25 nm of an MVR event during the observation period.

## Synapse identification and analysis of vesicle diameter in LaSEM data
Three characteristic features were used for synapse identification: the presence of a synaptic vesicle cluster, the postsynaptic density, and the uniform gap between pre- and postsynaptic membranes. Distance to the AZ center was determined from the projection of the vesicle center position to the AZ plane.

## Fit regression models
Nonlinear and linear fit regression models were generated using built-in functions in Matlab.

## Data inclusion and exclusion criteria

A minimum of five detected release events per bouton was required for all analyses.

## Statistical analysis

Statistical analyses were performed in Matlab. Statistical significance was determined using two-sample two-tailed t-test, paired t-test, Kolmogorov-Smirnov (K-S) test, one-way analysis of covariance (ANOCOVA) or chi-square test, where appropriate. The number of experiments reported reflects the number of different cell cultures tested. The value of N is provided in the corresponding figure legends and in *Table 1*. The statistical tests used to measure significance are indicated in each figure legend along with the corresponding significance level (p value). Data are reported as mean ± SEM and $p < 0.05$ was considered statistically significant. Analysis of the samples was not blinded to condition. Randomization and sample size determination strategies are not applicable to this study and were not performed.

# Acknowledgements

This work was supported in part by the R35 NS111596 grant to VAK from NINDS. We thank Drs Meyer Jackson and Gabrielle Edgerton for helpful comments on the manuscript.

# Additional information

### Funding

| Funder | Grant reference number | Author |
| --- | --- | --- |
| National Institute of Neurological Disorders and Stroke | NS111596 | Vitaly Klyachko |

The funders had no role in study design, data collection and interpretation, or the decision to submit the work for publication.

### Author contributions

Dario Maschi, Conceptualization, Data curation, Software, Formal analysis, Validation, Investigation, Visualization, Methodology, Writing - original draft, Writing - review and editing; Vitaly A Klyachko, Conceptualization, Resources, Software, Supervision, Funding acquisition, Methodology, Writing - original draft, Project administration, Writing - review and editing

### Author ORCIDs

Vitaly A Klyachko (iD) https://orcid.org/0000-0003-3449-243X

### Ethics

Animal experimentation: All animal procedures conformed to the guidelines approved by the Washington University Animal Studies Committee (protocol approval # 20170233).

### Decision letter and Author response

Decision letter https://doi.org/10.7554/eLife.55210.sa1
Author response https://doi.org/10.7554/eLife.55210.sa2

# Additional files

### Supplementary files

• Transparent reporting form

### Data availability

All data generated or analysed during this study are included in the manuscript and supporting files.

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
