## [Decision Letter]

**Acceptance summary:**

This paper presents a clever high-resolution imaging approach to measure spatial aspects of vesicles release at central synapses. Using a deconvolution approach the authors can distinguish single-vesicle and multi-vesicle release events. The key finding is that release events are inhomogeneously distributed across the active zone. Multi-vesicle events share release sites with single-vesicle events, and are more frequent at the center of the AZ, where the release probability is highest. This study provides an unprecedented view of spatial aspects of synaptic vesicles release.

**Decision letter after peer review:**

[Editors’ note: the authors submitted for reconsideration following the decision after peer review. What follows is the decision letter after the first round of review.]

Thank you for submitting your work entitled "Spatiotemporal dynamics of multi-vesicular release is determined by heterogeneity of release sites in central synapses" for consideration by *eLife*. Your article has been reviewed by two peer reviewers, and the evaluation has been overseen by a Reviewing Editor and a Senior Editor. The reviewers have opted to remain anonymous.

Our decision has been reached after consultation between the reviewers. Based on these discussions, we regret to inform you that your work will not be considered further for publication in *eLife*.

The reviewers and editor were excited by the approach and some of the data. However, the study is at a relatively preliminary stage and it is unlikely that the additional data and analysis required can be accomplished in two months. The *eLife* policy is to reject a paper under these conditions.

The mechanisms of release have often been studied with electrophysiological methods. However, these methods average over synapses, washing out interesting fluctuations that are informative about the function of single synapses. Over the last two decades a variety of postsynaptic and presynaptic imaging methods have been developed to look at synaptic transmission at single synapses. Some methods have high SNR and can be used for studying synapses in intact tissue (i.e. calcium imaging), whereas others (e.g. synaptopHluorin) promise better time resolution and spatial resolution, but are typically used in reduced preparations. The current study uses imaging synaptopHluorin to track single release events at individual synapses. The paper reports multi-vesicular release, with the dominant release site closer to the center of the active zone (AZ). Release probability was highest at the center of the AZ and dropped off towards the edge. This gradient, and spatial features of MVR, were similarly tightened by buffering intracellular calcium.

This is a nice imaging paper on vesicle release at single synapses. All reviewers agreed that the study is potentially very interesting, but don't find the data set and quantification convincing in their current state.

Essential revisions:

1) The paper has to be reframed. The Introduction might have been appropriate in the late 1990's, when UVR ("one bouton, one spike, one quantum") was a reasonable hypothesis based on prior work. Obviously, UVR was interesting in principle because it would pose serious biophysical riddles. If one vesicle releases, what signal tells other docked vesicles not to release? This signal would have to travel over microsecond timescales. The UVR came from work on the NMJ (misunderstood in this context) and, famously, the Mauthner cell. The original data has been reanalyzed and the statistical analysis has been questioned (some would say debunked, see Ninio, 2007, J. Neurophysiol.).

At central synapses support for UVR was always very spotty. In contrast, imaging studies with single synapse resolution tended to show MVR, as did many excellent electrophysiological studies using more indirect analyses. Yes, individual synapses often release single quanta, but this is because their overall release probability is often low. Under conditions of elevated release probability quanta are released based on the binomial model (e.g. Oertner et al., 2002; reviewed in Rudolph et al., 2015). It seems there is no evidence for UVR left and the whole thing is best forgotten except as a lesson in science sociology.

This means that MVR and UVR are likely the same process (operating independently at release sites/docked vesicles with different individual release probs). The authors have demonstrated that sites that host MVR could also harbor a UVR event later. This interchangeability actually suggests that they may be similarly regulated. In fact, none of the reviewers see support for a difference in spatial or temporal profile between MVR and UVR. Please analyze the distributions of time intervals for a UVR occurring at sites of an MVR, an MVR occurring at sites of an MVR, a UVR occurring at sites of a UVR, and a UVR occurring at sites of a UVR to see any difference. In Figure 1I the authors did not classify UVR into distal UVR and proximal UVR, as they did for MVR. This may have biased the comparison.

2) The authors concluded that they observed desynchronization of MVR events based on the ~8-10% difference in fluorescence intensity profiles of doublet events. However, from Figure 3A, the difference in the peak fluorescence intensities of two events is surely larger than 25%. Given 40 ms exposure time, it is hard to believe that ~4 ms of desynchronization generated such a large difference in fluorescence intensities. Moreover, if it is truly due to desynchronization, we would expect that the fluorescence intensity profiles of two events reversed over the next time point. Please explain.

3) Related to above. The second main finding is a desynchronization of MVR, with the more central release site leading and the peripheral site lagging by up to 4 ms. This evidence is indirect (via amplitude) and somewhat unconvincing (see above). How do we know that the amplitude difference is due to timing and not due to other factors (cleft pH, out-of-focus release, vesicle size etc.)?

As discussed in the 2017 Neuron paper, AZs may be tilted relative to the imaging plane, which could also affect the measured amplitude of distal events. Given these alternative explanations, it would be reassuring to have confirmatory evidence for delayed release events. A 4 ms delay is huge, twice the typical synaptic delay. Such delayed events, even if rare, should be readily visible in voltage clamp recordings. This is all a bit puzzling and needs to be explained better.

4) The authors argue that EGTA "affected the MVR event de-synchronization" and abolishes "the preferential localization of the earlier event in the MVR pair closer to the AZ center". In both cases, a significant difference under control conditions (Figure 3E, Figure 3C) was found 'non-significant' in EGTA (Figure 4B, Figure 4D). However, the number of analyzed synapses was more than twice as high under control conditions, which could account for the higher significance found in the t-tests. To compare t-tests of identical power, it may be necessary to conduct additional EGTA experiments. More importantly, the correct comparison is testing for changes in the difference between smaller and larger with and without EGTA. As is, the analysis is statistically flawed. This likely requires more experiments and analysis.

[Editors’ note: further revisions were suggested prior to acceptance, as described below.]

Thank you for choosing to send your work entitled "Spatiotemporal dynamics of multi-vesicular release is determined by heterogeneity of release sites in central synapses" for consideration at *eLife*.

Specifically, while the reviewers and editors found the manuscript much improved, they felt that a complete rewrite would still be necessary, and therefore this version still falls short of what is needed. One reviewer stated:

“In my opinion, the results strongly suggest that active zones close to the center of a synapse have a higher release probability than more distal AZs. This is an important and interesting finding that should be simple to understand. Unfortunately, it is (still) obfuscated by their MVR/UVR site classification, which they now partially take back ("our results cannot be interpreted to indicate that there are some specialized release sites that only support MVR or UVR"). But the confusion already starts at Figure 1A, where a release site is labeled in red (for MVR) that has in fact released only a single vesicle. So while it makes sense to talk about UVR and MVR *events*, it is very confusing to talk about UVR and MVR *sites*, only to state at the end that they are probably not fundamentally different. (Since typically only a single MVR event was observed per bouton, meaningful single bouton statistics is not possible). What is needed is a complete rewrite of the manuscript, including a change in nomenclature and analysis strategy, not just the addition of some text blocks.”

---

## [Author Response]

Essential revisions:1) The paper has to be reframed. The Introduction might have been appropriate in the late 1990's, when UVR ("one bouton, one spike, one quantum") was a reasonable hypothesis based on prior work. Obviously, UVR was interesting in principle because it would pose serious biophysical riddles. If one vesicle releases, what signal tells other docked vesicles not to release? This signal would have to travel over microsecond timescales. The UVR came from work on the NMJ (misunderstood in this context) and, famously, the Mauthner cell. The original data has been reanalyzed and the statistical analysis has been questioned (some would say debunked, see Ninio, 2007, J. Neurophysiol.).At central synapses support for UVR was always very spotty. In contrast, imaging studies with single synapse resolution tended to show MVR, as did many excellent electrophysiological studies using more indirect analyses. Yes, individual synapses often release single quanta, but this is because their overall release probability is often low. Under conditions of elevated release probability quanta are released based on the binomial model (e.g. Oertner et al., 2002; reviewed in Rudolph et al., 2015). It seems there is no evidence for UVR left and the whole thing is best forgotten except as a lesson in science sociology.

We completely agree, and have extensively reframed the manuscript to reflect this point. We now state in the Introduction that:

“Although initially it was hypothesized that, at most, only a single vesicle can be released from a given synapse with each action potential (i.e. uni-vesicular release (UVR)), it is now widely accepted that two or more vesicles can fuse simultaneously in response to a single action potential in the same synaptic bouton, leading to the notion of multi-vesicular release (MVR) (Rudolph et al., 2015). […] Despite its prevalence, spatiotemporal organization of MVR within the synaptic active zone (AZ) and its regulation are poorly understood (Rudolph et al., 2015)”.

This means that MVR and UVR are likely the same process (operating independently at release sites/docked vesicles with different individual release probs). The authors have demonstrated that sites that host MVR could also harbor a UVR event later. This interchangeability actually suggests that they may be similarly regulated. In fact, none of the reviewers see support for a difference in spatial or temporal profile between MVR and UVR. Please analyze the distributions of time intervals for a UVR occurring at sites of an MVR, an MVR occurring at sites of an MVR, a UVR occurring at sites of a UVR, and a UVR occurring at sites of a UVR to see any difference. In Figure 1I the authors did not classify UVR into distal UVR and proximal UVR, as they did for MVR. This may have biased the comparison.

We thank the reviewers for this important point and agree. We note that this concern is caused in a large part by a misunderstanding due to lack of clarity in the original version of our manuscript. We did indeed classify UVR into distal and proximal in Figure 1I, as we did for MVR, but we failed to note this clearly in the figure. We now clarify that this analysis was performed on MVR and UVR events equidistant to the AZ center (Figure 1H). In other words, for each MVR event pair, the proximal MVR site was compared only to the proximal UVR sites, and the distal MVR site was compared only to the distal UVR sites. These results show that the release sites that are more likely to harbour both UVR and MVR are characterized by higher release probability than other equidistant sites.

We also completely agree with the reviewers that our results do not indicate that there are specialized sites of UVR vs. MVR, but rather the same sites can harbor both types of events. In fact, our finding that MVR sites have higher release probability is based on observation that UVR events occur more frequently at the release sites at which MVR events are observed. We have now expanded the description of the results to clarify this point. We also stated in the Results that “our results cannot be interpreted to indicate that there are some specialized release sites that *only* support MVR or UVR”.

Following the reviewers’ suggestion, we also attempted to compare the distributions of time intervals of UVR and MVR events occurring at sites of UVR or MVR and vice versa. However, we came to realization that this analysis is not possible within our current approach because we cannot determine the distribution of time intervals between MVR events at any given release site. This is because the very low probability of observing MVR events, which is typically only 1-2 per bouton within the 120 sec observation window. As a result, MVR events were almost never observed at the same release site more than once in the same bouton during this time period, thus making analysis of inter-event intervals not feasible. We previously showed that intrinsic slow displacement of synaptic boutons precludes precise analysis of release event localization for much longer time periods, which is what would be required for such analysis.

In summary, we have revised the manuscript to include these points and to clarify that our data does not suggest that there are specialized sites of MVR vs UVR, but rather the same sites support both types of release. We further emphasize that it was not the intent of our study to contrast UVR vs MVR. Rather we believe that the critical finding is the highly heterogeneous properties of release sites within the individual AZs. The sites with higher release probability are located closer to the AZ center and are more likely to harbor both UVR and MVR events. The spatiotemporal features of MVR reflect this heterogeneity of release sites across individual AZs and depend, in part, on the non-uniform landscape of calcium elevation across the AZ following an action potential. Our analyses thus suggest a new level of structural and functional organization of release sites within individual AZs that determines the spatiotemporal dynamics of MVR. We have also revised the Introduction and Discussion of the manuscript to clarify these points.

2) The authors concluded that they observed desynchronization of MVR events based on the ~8-10% difference in fluorescence intensity profiles of doublet events. However, from Figure 3A, the difference in the peak fluorescence intensities of two events is surely larger than 25%. Given 40 ms exposure time, it is hard to believe that ~4 ms of desynchronization generated such a large difference in fluorescence intensities. Moreover, if it is truly due to desynchronization, we would expect that the fluorescence intensity profiles of two events reversed over the next time point. Please explain.

We thank the reviewers for these points and performed extensive additional experiments and analyses to determine origins of amplitude differences within MVR events, and to solidify our interpretation of these differences as evidence for desynchronization of MVR. Specifically:

a) Following the reviewer’s suggestions, we perform additional studies to demonstrate that amplitude differences within MVR events are not due to differences in vesicle size (Figure 3—figure supplement 1D) or changes in cleft pH (Figure 3—figure supplement 1B). We also clarified in the text that due to the configuration of our experiments, the observed differences cannot arise simply from the tilt of AZs relative to the imaging plane. This is also evident in the observation that the average amplitude of UVR events is indistinguishable at the AZ center vs periphery (Figure 3—figure supplement 1B, insert). These results are discussed in more detail in the response to point #3 below.

b) We performed extensive additional experiments to confirm that the observed scaling of the amplitude difference within the MVR events as a function of the intra-event separation is eliminated by EGTA-AM (and see response to point #4 below for more details).

c) We performed additional analyses to examine how much the uncertainty in determination of fusion event amplitude contributes to the observed amplitude differences within the MVR event pairs. We found that this uncertainty is comparable to the amplitude differences of the closely-spaced MVR events (~10%), but cannot account for the amplitude differences within MVR events with large separation (Figure 3—figure supplement 1A). Moreover, the positive correlation between the spatial separation and amplitude difference of the two events comprising MVR cannot be explained by measurement uncertainty.

d) We clarified in the text that desynchronization of MVR events is a well established phenomenon, with several studies reporting MVR event desynchronization in the range of ~1-5ms in both excitatory and inhibitory central synapses (Auger et al., 1998; Auger and Marty, 2000; Crowley et al., 2007; Malagon et al., 2016; Rudolph et al., 2011). We also emphasized that our initial estimate for the maximal time difference within our double events of ~4ms is an overestimate because a component of the amplitude differences arise from measurement uncertainty. We provided a corrected upper bound estimate on the MVR desynchronization to be ~2ms, if the measurement uncertainty is factored in. These values are well within the range of 1-5ms reported previously for MVR event desynchronization in the above studies.

e) The reviewers pointed out that the fluorescence intensity profiles of two events are expected to reverse over the next time point if the differences are truly due to desynchronization. Indeed, we agree that this would be the case, but only if the dwell time of the fluorescent signal generated by the individual events is exactly the same for both events. However, unlike the precise initiation of fusion, the dwell time of individual events is highly variable, spanning from ultrafast (~150-250 ms), to fast (~5-12 s) to ultraslow (>20 s) (Chanaday and Kavalali, 2018). A few initial milliseconds of desynchronization do not have any significant impact over such variability. Therefore there is no correspondence between the timing of the event start in one frame and the duration of observation of the same event in any subsequent frame.

f) The example shown in Figure 3A has a ~25% difference in amplitude within the MVR event and was chosen to highlight the differences in amplitude, which can be visually obscured by pixelation and noise. This example is reasonably representative of the differences observed for the far apart MVR events, but we can replace it with a different example if reviewers suggest it.

3) Related to above. The second main finding is a desynchronization of MVR, with the more central release site leading and the peripheral site lagging by up to 4 ms. This evidence is indirect (via amplitude) and somewhat unconvincing (see above). How do we know that the amplitude difference is due to timing and not due to other factors (cleft pH, out-of-focus release, vesicle size etc.)?As discussed in the 2017 Neuron paper, AZs may be tilted relative to the imaging plane, which could also affect the measured amplitude of distal events. Given these alternative explanations, it would be reassuring to have confirmatory evidence for delayed release events. A 4 ms delay is huge, twice the typical synaptic delay.

We thank the reviewers for this important point. As mentioned in the response to point #2 above, we performed extensive additional experiments to address the reviewer’s concerns. We provide more details of these experiments here:

a) Following the reviewers’ suggestions, we examined if amplitude differences within MVR events can arise from differences in vesicle size. We determine the vesicle size from the EM microscopy of our synapses in culture as a function of distance to the AZ center (Figure 3—figure supplement 1C, D). Vesicle size did not exhibit significant changes with distance to the AZ center (Figure 3—figure supplement 1D). Thus the amplitude differences within MVR events are not due to differences in vesicle size.

b) We also determined if amplitude differences within MVR events can arise from differences in the cleft pH at different locales of the AZ. If this is the case, changes in the cleft pH should similarly affect amplitudes of UVR and MVR events. Yet we did not detect any significant changes in the intensity of UVR events throughout the whole AZ (Figure 3—figure supplement 1B). Thus changes in cleft pH are unlikely to explain the differences in MVR event amplitude.

c) We clarified in the text that in our recordings dozens of synapses are positioned in random orientation and are recorded simultaneously. Some boutons have the center of the AZ in focus and others have the periphery, and yet others are positioned in all other possible configurations inbetween (schematic in Author response image 1). Thus there is no any systematic bias towards any one possible configuration or tilt of the AZ relative to the imaging plane that can lead to systematically larger fusion event amplitude at the AZ center vs. periphery. Indeed, this notion is highlighted by the fact that the average UVR event amplitude is invariable in our measurements, independently of the distance to the AZ center (Figure 3—figure supplement 1B, insert). Thus the observed differences in MVR event amplitude are unlikely to result from tilt of AZ relative to the imaging plane.

**Author response image 1. respfig1:** Schematic of the wide range of possible AZ positions relative to the focal plane.

Such delayed events, even if rare, should be readily visible in voltage clamp recordings. This is all a bit puzzling and needs to be explained better.

We completely agree and would like to emphasize that desynchronization of MVR events have indeed been reported previously in a number of voltage clamp studies in both excitatory and inhibitory simple synapses:

a) Widespread desynchronization of MVR with delays in the range of 1-5ms within the event pairs was found at single synapses formed by parallel fiber to interneuron connections in cerebellar slices (Figure 6B in Malagon et al., 2016), with examples of up to 5-8ms delays within MVR events (Figure 2B, 3A, 5A in Malagon et al., 2016).

b) Desynchronization of MVR events in the range of 1-10ms was reported at single granule cell to stellate cell synapses in rat cerebellar slices (Figure 5B,E in Crowley et al., 2007).

c) Jitter in the timing within MVR events on the millisecond timescale was found at climbing fiber to Purkinje cell synapses under similar stimulation conditions to ours (Rudolph et al., 2011).

d) Widespread desynchronization of MVR events in the range of 1.5-3.5ms was reported at single synaptic connections between cerebellar stellate and basket cells (Auger et al., 1998; Auger and Marty, 2000).

To address the reviewer’s concern, we have cited these studies and clarified this point in the Results of the revised version. We further emphasize that the novel observation in our study is not in the fact that MVR can be desynchronized, but in revealing a pattern of spatial organization within these MVR events relative to the AZ center, that is correlated with the gradient of release site properties and is regulated by calcium elevation.

4) The authors argue that EGTA "affected the MVR event de-synchronization" and abolishes "the preferential localization of the earlier event in the MVR pair closer to the AZ center". In both cases, a significant difference under control conditions (Figure 3E, Figure 3C) was found 'non-significant' in EGTA (Figure 4B, Figure 4D). However, the number of analyzed synapses was more than twice as high under control conditions, which could account for the higher significance found in the t-tests. To compare t-tests of identical power, it may be necessary to conduct additional EGTA experiments. More importantly, the correct comparison is testing for changes in the difference between smaller and larger with and without EGTA. As is, the analysis is statistically flawed. This likely requires more experiments and analysis.

We thank the reviewers for this important point and absolutely agree. We performed the suggested additional experiments and have revised our statistical analyses as suggested. Specifically, the number of analyzed synapses is now 225 and 245 from 11 independent cultures each, for measurements with and without EGTA, respectively. We also corrected our initial statistical analyses and tested for changes in the slope of the dependence of the amplitude difference between smaller and larger events with and without EGTA, as reviewers suggested. We found that EGTA indeed significantly reduced it (Figure 4B). Similarly, we still observed that proportion of MVR events that had shorter intra-event separation increased significantly in the presence of EGTA (Figure 4C). Accordingly, the average distances from both events in the MVR pair to the AZ center were significantly shortened in the presence of EGTA (Figure 4—figure supplement 1A). Thus calcium buffering causes MVR events to occur at shorter separation distances and more proximal to the AZ center. However, the larger event in the MVR pair was still localized closer to the AZ center in the presence of EGTA than the smaller one in the pair (Figure 4D). This is consistent with the observation that release sites with higher release probability are still localized closer to the AZ center in the presence of EGTA (Figure 4E) and that preferential utilization of more central release sites was exacerbated in the presence of EGTA (Figure 4E). We have revised Figure 4 and the description of these results accordingly.

[Editors’ note: what follows is the authors’ response to the second round of review.]

Specifically, while the reviewers and editors found the manuscript much improved, they felt that a complete rewrite would still be necessary, and therefore this version still falls short of what is needed. One reviewer stated:“In my opinion, the results strongly suggest that active zones close to the center of a synapse have a higher release probability than more distal AZs. This is an important and interesting finding that should be simple to understand. Unfortunately, it is (still) obfuscated by their MVR/UVR site classification, which they now partially take back ("our results cannot be interpreted to indicate that there are some specialized release sites that only support MVR or UVR"). But the confusion already starts at Figure 1A, where a release site is labeled in red (for MVR) that has in fact released only a single vesicle. So while it makes sense to talk about UVR and MVR events, it is very confusing to talk about UVR and MVR sites, only to state at the end that they are probably not fundamentally different. (Since typically only a single MVR event was observed per bouton, meaningful single bouton statistics is not possible). What is needed is a complete rewrite of the manuscript, including a change in nomenclature and analysis strategy, not just the addition of some text blocks.”

We thank the reviewer for this important point and fully agree. We made major revisions to the text and figures of the manuscript to address the reviewer’s criticism as follows:

1) We would like to clarify that the reviewer’s concern was in a large part caused by the lack of clarity in our description of the results, but not in the actual analyses themselves. Specifically, we did not classify sites into UVR or MVR sites in any of the analyses presented in the manuscript, but our description made an unintended impression that this was the case. The vast majority of analyses, except for Figures 1G-I and Figure 4E-G, examined individual UVR and MVR release events only. In the remaining analyses, we compared release probability of release sites that did or did not have an MVR event during the observation period. These analyses were based on counting all release events at each release site and did not distinguish UVR vs. MVR events in the calculation of P_r_-site (see below for more details). We have improved the schematics of these measurements in the figures and the presentation of these results in the text to clarify this point.

2) We changed the nomenclature used in the manuscript as the reviewer suggested. Only the terms “UVR events” and “MVR events” are now used throughout the manuscript. We completely remove any mention of “UVR release sites” or “MVR release sites”. We now use the same generic term “release site” for both forms of release throughout the manuscript.

3) We improved the presentation of UVR and MVR event measurements in Figure 1A to clarify that all release sites were considered in the same way, no matter whether the release was a UVR event or an MVR event, as the reviewer suggested.

4) We improved the schematics of the P_r_-site measurements in Figure 1H and 4G, and the corresponding legends and the bar graphs in the same panels, to clarify that release sites with or without an MVR event were compared. We further clarify in the Materials and methods that calculation of P_r_-site did not distinguish UVR vs MVR events. Each of the two release events comprising an MVR pair was counted independently as a single release event at the two release sites that harbored them.

5) Following the reviewer’s suggestion we extensively revised the Summary, Introduction, Results and Discussion of the manuscript to highlight the critical finding of a gradient of release site release probability from the AZ center to periphery.

6) We extensively revised the manuscript to improve clarity of result presentation and discussion. In summary, following the reviewer’s criticism, we have extensively revised the manuscript to clarify that our study distinguishes only UVR events and MVR events, but we do not analyze or distinguish release sites as “UVR sites” or “MVR sites”.

**References:**

Auger, C., Kondo, S., and Marty, A. (1998). Multivesicular release at single functional synaptic sites in cerebellar stellate and basket cells. J Neurosci *18*, 4532-4547.

Auger, C., and Marty, A. (2000). Quantal currents at single-site central synapses. J Physiol *526 Pt 1*, 3- 11.

Chanaday, N.L., and Kavalali, E.T. (2018). Optical detection of three modes of endocytosis at hippocampal synapses. *eLife 7*.

Crowley, J.J., Carter, A.G., and Regehr, W.G. (2007). Fast vesicle replenishment and rapid recovery from desensitization at a single synaptic release site. J Neurosci *27*, 5448-5460.

Foster, K.A., Crowley, J.J., and Regehr, W.G. (2005). The influence of multivesicular release and postsynaptic receptor saturation on transmission at granule cell to Purkinje cell synapses. J Neurosci *25*, 11655-11665.

Malagon, G., Miki, T., Llano, I., Neher, E., and Marty, A. (2016). Counting Vesicular Release Events Reveals Binomial Release Statistics at Single Glutamatergic Synapses. J Neurosci *36*, 4010-4025.

Oertner, T.G., Sabatini, B.L., Nimchinsky, E.A., and Svoboda, K. (2002). Facilitation at single synapses probed with optical quantal analysis. Nat Neurosci *5*, 657-664.

Rudolph, S., Overstreet-Wadiche, L., and Wadiche, J.I. (2011). Desynchronization of multivesicular release enhances Purkinje cell output. Neuron *70*, 991-1004.

Wadiche, J.I., and Jahr, C.E. (2001). Multivesicular release at climbing fiber-Purkinje cell synapses. Neuron *32*, 301-313.